# Influenza Vaccination Practices and Perceptions Among Young Athletes: A Cross-Sectional Study in Greece

**DOI:** 10.3390/vaccines12080904

**Published:** 2024-08-09

**Authors:** Dimitrios Lamprinos, Maria Vroulou, Michail Chatzopoulos, Panagiotis Georgakopoulos, Paraskevi Deligiorgi, Evangelos Oikonomou, Gerasimos Siasos, Petros G. Botonis, Kostas A. Papavassiliou, Dimitrios Papagiannis, Theodoros Pouletidis, Christos Damaskos, George Rachiotis, Georgios Marinos

**Affiliations:** 1Emergency Care Department, Laiko General Hospital, 11527 Athens, Greece; dimitrislamprinos@gmail.com (D.L.); maraki_vrl@hotmail.gr (M.V.); mi.chatzopoulos@gmail.com (M.C.); panos.k.georgakopoulos@gmail.com (P.G.); 2First Department of Cardiology, Hippokration General Hospital, Medical School, National and Kapodistrian University of Athens, 11527 Athens, Greece; evi_deligiorgi@hotmail.com (P.D.); boikono@gmail.com (E.O.); gsiasos@med.uoa.gr (G.S.); tpouletidis@gmail.com (T.P.); 3Third Department of Cardiology, Thoracic Diseases General Hospital Sotiria, Medical School, National and Kapodistrian University of Athens, 11527 Athens, Greece; 4Department of Sports Medicine and Biology of Exercise, Faculty of Physical Education and Sport Science, National and Kapodistrian University of Athens, 17237 Athens, Greece; pboton@phed.uoa.gr; 5First Department of Respiratory Medicine, “Sotiria” Hospital, Medical School, National and Kapodistrian University of Athens, 11527 Athens, Greece; konpapav@med.uoa.gr; 6Public Health & Vaccines Laboratory, Department of Nursing, School of Health Science, University of Thessaly, 41500 Volos, Greece; dpapajon@gmail.com; 7N.S. Christeas Laboratory of Experimental Surgery and Surgical Research, Medical School, National and Kapodistrian University of Athens, 11527 Athens, Greece; x_damaskos@yahoo.gr; 8Department of Hygiene and Epidemiology, Faculty of Medicine, University of Thessaly, 41500 Larissa, Greece; grachiotis@gmail.com; 9Department of Hygiene, Epidemiology and Medical Statistics, School of Medicine, National and Kapodistrian University of Athens, 11527 Athens, Greece

**Keywords:** vaccine, sports, preventive medicine

## Abstract

Background: Influenza vaccination among athletes is a crucial area in sports medicine. This descriptive, cross-sectional study aims to explore the vaccination practices and intentions regarding influenza vaccines among young athletes. Methods: A structured, questionnaire-based study was conducted among students from the National School of Sports in Greece. The survey was conducted over the period of April to May 2023. Overall, 138 participants participated in the study. Results: More than half of the participants had received a flu vaccine in the past, but only 12.3% were vaccinated against influenza for 2022–2023. The main reasons seemed to be the lack of time (40.6%) and the idea that influenza does not lead to any serious health threats for the participants (36.2%). The main factor that affected their decision to get the flu vaccine or not was the need for more information regarding influenza vaccination (79%). Conclusions: The recent study showed low vaccination coverage among people of young age participating in sports activities. The qualitative views of the participants highlighted the significance of the lackof a well-organized information program provided by health professionals and coaches.

## 1. Introduction

Influenza, commonly known as the flu, is the most frequent cause of acute respiratory disease requiring medical intervention affecting all age groups. Influenza viruses (influenza viruses A, B, and C and thogotovirus) are enveloped particles with two surface glycoproteins—haemagglutinin and neuraminidase. The combination of different subtypes leads to antigenic variation. The ability of these viruses to undergo antigenic variation by the combination of different subtypes often leads to the emergence of novel strains that can trigger outbreaks when person-to-person transmission (airborne) occurs within large, immunologically susceptible populations. Influenza in adults and adolescents typically presents with an abrupt onset of fever and chills, accompanied by headache, sore throat, myalgias, malaise, anorexia, and a dry cough [1]. Those symptoms can severely impact an individual’s daily life and well-being, but for athletes, these effects can extend further, affecting their performance and training programs. Even mild cases of influenza can significantly reduce an athlete’s ability to engage in training and competition [2].

In the context of sports, athletes face unique challenges when it comes to preventing influenza infection. Prevention of influenza infection is a key issue in the health care of athletes. Vaccination, personal hygiene measures, social distancing, and avoiding people with flu-like symptoms play major roles in these matters. However, some typical circumstances of athletes’ daily lives, such as frequent travelling and close contact with teammates and opponents, might make it challenging to implement these preventive measures and indicate the need for a well-organized influenza vaccination schedule. It has been pointed out that even mild diseases could have a negative impact on the performance of athletes. Moreover, even minor infections can reduce the athlete’s ability to sustain heavy training. In addition, the intense physical activity of training and competition, with its possible effects on the immune function, can potentially influence decisions about the execution and timing of vaccination [3,4]. Vaccine guidelines target mainly public health issues and focus on the general population rather than on individuals with different benefit–risk profiles. Currently, licensed influenza vaccines are trivalent-inactivated formulations that contain each of the haemagglutinins of the influenza A (H1N1), influenza A (H3N2), and influenza B strains. The recommended composition of the influenza virus vaccine is updated annually to provide vaccines antigenically well-matched with the new influenza virus strains that are expected to cause epidemics. Therefore, the administration of the influenza vaccine every year before the predicted epidemic period can significantly reduce the number of cases of flu infections and avoid complications in high-risk groups [1]. According to the National Greek Vaccination Guidelines of 2023, people over 65 years of age and people with a high risk of infection, like medical personnel, or severe diseases should be vaccinated against influenza viruses. The influenza vaccine is fully covered financially by the government, ensuring that cost is not a barrier for those in the recommended groups. For athletes and others outside of these specific groups, access to vaccination is the same with the abovementioned groups, as they can visit their family doctor for a medical prescription, and the vaccine can be administered by pharmacists. Although, there is no specific mention of people participating in sports, a special group that we consider likely to belong to the high-risk groups for disease and the transmission of the viruses [5]. Recent literature, particularly from during the COVID-19 pandemic, has highlighted the importance of effective communication about vaccination. Cervi et al. (2023) analyzed digital communication strategies and found that clear and accessible information significantly influences public health behaviors. Understanding how young athletes receive their information about health and vaccination is crucial in addressing vaccination hesitancy [6]. Our study aims to explore the vaccination practices and intentions regarding influenza vaccines in athletes. Our findings may contribute to the improvement of the overall health and athletic performance of athletes, allowing them to compete in their respective sports while safeguarding their overall well-being.

## 2. Materials and Methods

Our study uses a descriptive research design, utilizing a simple univariate analysis approach. Data collection was conducted through an online survey distributed among students studying physical education and sport science in Greece. The survey was conducted over the period of April to May 2023. All students were invited to participate in the anonymous survey. A structured, anonymous questionnaire was used. The study adhered to the principles of the Declaration of Helsinki of 1975, as revised in 2008. Participants provided anonymous informed consent on the survey platform before proceeding to complete the questionnaire. The protocol of the study was approved by the N.S. Christeas Laboratory of Experimental Surgery and Surgical Research (protocol number: 12/02-12-2022). We invited 250 athletes, and 138 of them agreed to participate in the study (response rate 55.2%). The questionnaire included questions on demographics (sex, age, sport, and level of activity) and perceptions of the importance of vaccinations and the safety and effectiveness of vaccines. Respondents were subdivided into three categories based on their level of athletic involvement: amateur, semi-professional, and professional athletes. This categorization was determined according to their participation in championships at local, national, and international levels. Specifically, amateur athletes participated in local competitions, semi-professional athletes competed at the national level, and professional athletes were involved in international championships. Additionally, the questionnaire included questions about influenza vaccination history (“Have you been vaccinated against influenza?” [answer options: yes/no]) and influenza vaccination coverage for the 2022–23 flu season: “Have you been vaccinated with the influenza vaccine (season 2022–2023)?” (answer options: yes/no). In the case of vaccination refusal, the participants were requested to report the reason for non-vaccination (answer options: “fear of the procedure”, “I did not have time”, “I am not in danger from influenza”, “I believe that I will have flu symptoms from the vaccine”, and “fear over side effects”). The participants were also asked about possible motivations for vaccine uptake (answer options: “more information”, “economic reward”, and “easier access”). The participants rated the importance, effectiveness, and safety of vaccinations on a four-point Likert scale (answer options: “fully agree”, “agree”, “disagree”, and “fully disagree”). They were also asked about their sources of information on influenza vaccination. Finally, information about the effects of the influenza vaccine on their training and official game participation was obtained (“Do you believe that you are going to lose training days due to the vaccine side effects?” [answer options: yes/no]; “Have you lost training days due to the vaccine side effects?” [answer options: yes/no]; “Do you believe that you are going to lose an official game due to the vaccine side effects?” [answer options: yes/no]; and “Have you ever lost an official game due to the vaccine side effects?” [answer options: yes/no]). The collected data were analyzed using IBM SPSS Statistics software (SPSS 23). Descriptive statistics were used to summarize the demographic characteristics of the respondents and their vaccination status. To examine associations between the categorical variables (e.g., vaccination status and various demographic/sport-related factors), chi-square tests (χ^2^) were conducted. Additionally, independent *t*-tests were employed to compare means between the vaccinated and non-vaccinated groups for continuous variables. Differences were considered as statistically significant at a *p*-value < 0.05.

## 3. Results

A total of 138 students agreed to participate in our study. Approximately 58% (n = 80) of the participants were females. The mean age was 21.8 years (SD = 3.96, min = 18 max = 43); 59.4% (n = 82) were amateur-level, 29.7% (n = 41) were semi-pro-level, and 10.9% (n = 15) were pro-level athletes. Almost one-fourth of the responders were water sports athletes (23.9%, n = 33), 18.8% (n = 26) played team sports, 17.4% (n = 24) played field sports, 13% (n = 18) played fighting sports, and the rest, 26.8%, were categorized as others, including tennis, dancing, weightlifting, and gymnastics (Table 1). More than half of the participants (55.8%, n = 77) had taken a flu vaccine in the past, but 87.7% (n= 121) did not uptake the flu vaccine for 2022–2023. Almost half (51.4%, n = 71) had the belief that the vaccine could cause flu infection. The main reasons for not receiving the flu vaccine were lack of time (40.6%, n = 56), fearless over flu infection (36.2%, n = 50), fear over vaccine side effects (7.2%, n = 10), and the belief that the vaccine could cause flu-like symptoms (3.6%, n = 5). Approximately 12.3% (n = 17) were not willing to mention their reasons for no vaccination (Table 2).

The factors that could affect their decision to obtain the flu vaccine or not were more information regarding influenza vaccination (79%, n = 109), economic rewards (11.6%, n = 16), and easier access to the vaccine (9.4%, n = 13). Almost three-fourths of the participants who did not receive the vaccine for the 2022–2023 flu season were informed by social media, media, and independent websites (72.5%, n = 100). More than half of the responders (53.6%, n = 74) said they would suggest flu vaccination to their co-athletes. Lastly, 73.9% (n = 102) responded that they needed more information about vaccination against influenza. The factors that we found to be significantly associated with the decision not to be vaccinated for the 2023 flu season were the thought that vaccines are generally safe and the lack of fear that the side effects of the vaccine could lead to a loss of an official game. Most of the responders with the belief that vaccines are important for public health did not receive the vaccine for 2022–2023, but we did not find a significant association (88.5%, *p* = 0.179). Also, no association was found with age, gender, the sports category, or the level of training (Table 3).

As a secondary analysis, we tried to find associated factors with the approval of the flu vaccine in general. Belief in not losing an official game due to vaccine side effects and history of influenza infection seemed to be positively associated with vaccine uptake (*p* = 0.001 and *p* = 0.002, respectively) (Table 4). No other factor was found to be significantly associated with influenza vaccination in the past.

## 4. Discussion

Highly contagious respiratory tract infections such as influenza can be easily transmitted among athletes, especially those participating in group sports that require close physical proximity [5]. Vaccination against influenza has been widely demonstrated as the main factor in preventing infection by the virus and the progression of the disease [7]. Our study and analysis demonstrated a strong hesitancy in young athletes toward the vaccine. Similar results have been reported in the past indicating that elite athletes avoid being vaccinated against influenza mainly because of the fear of the adverse effects of the vaccine and a lack of risk perception [8,9]. In 2022, G. Marinos et al. reported that the main reasons non-professional athletes avoided vaccination for COVID-19 were fear of vaccine safety, concerns about the short time period of the vaccine’s development and testing, and doubt about the risk of being exposed to SARS-CoV-2 [10]. Our study demonstrated that the main deterring factors seemed to be the lack of time (40.6%) and the idea that influenza is not a major health threat to the participants (36.2%). These findings are consistent with those of previous research on vaccine hesitancy for other vaccines and in different populations. Similar studies have reported that time constraints and perceived low risk of infection are significant barriers to vaccination for diseases perceived as relatively benign [8,11,12]. It is also notable that even though the great majority of the participants stated that vaccines in general are safe, effective, and important for public health, only 12.3% of the study subjects did uptake the vaccine for influenza for the current period (2022–2023). There is a connection between immune system modifications and physical activity. Thus, injuries and infections are among the many medical issues that athletes are vulnerable to. A significant medical condition that could be the cause of an athlete’s absence from training is infection. The connection between the immune system and physical activity, the characteristics of different infections in athletes, especially those with unique clinical presentations or complications, guidelines for when it’s safe to resume training, and strategies to prevent infections from developing and spreading among athletes [13]. Infection with influenza viruses can have serious and debilitating consequences for athletes. Even mild symptoms can lead to abstinence from training and reduced sports performance [14]. The vast majority of athletes are young. About fifteen years before, 2009 saw the first worldwide influenza virus pandemic since 1968, with younger people being infected more frequently. The influenza virus remains a threat for athletes, especially in close-contact sports [13,15]. Our analysis demonstrated it is more likely for young athletes who have previously been infected with influenza to be vaccinated against the virus (almost 68% of previously infected participants were vaccinated for the current period compared to almost 42% of the participants who had not contracted the virus previously). A logical interpretation of this observation is that the experience of a previous infection made the young athletes aware of the severe consequences influenza can have on their performance. Similar studies also showed that a higher perception of disease susceptibility and severity may contribute to a higher willingness to be vaccinated [11,16]. One of the most important components, a basic human right, and a responsibility of governments and other stakeholders as part of the 2030 Agenda for Sustainable Development and Its Sustainable Development Goals is access to a safe and healthy work environment [17,18]. The Centers for Disease Control and Prevention, World Health Organization, and Immunization Action Coalition make recommendations, emphasize that vaccination is the most effective way to protect against severe illness from infectious diseases, and strongly encourage athletes to get vaccinated [19]. These data suggest that the ultimate way to acquire higher vaccination coverage among people of young age participating in sport activities is through a well-organized information plan in which sports medical teams will play a crucial role. In 2020, Papagiannis et al. reported that the great majority (87%) of medical teams in Greece advise professional football players to be vaccinated for influenza [20]. Conversely, a notable proportion of vaccine hesitancy has been observed among physicians, which could impact vaccine acceptability [8]. Research indicates that concerns about the adverse effects of the influenza vaccine have led to a decline in public confidence [21,22]. In light of these findings, it is important to enhance efforts aimed at developing robust and effective immunization strategies. This involves instructing health care professionals to prioritize preventive medicine and the maintenance of good health, as well as implementing comprehensive educational and communication programs [23,24,25,26]. The testimonies of athletes who have been previously infected could also be used as a valuable tool to persuade others to be vaccinated. The need for a well-structured information program is also highlighted by the students themselves, as 79% of the participants required more information to decide whether they would undertake the vaccine. The importance of appropriate and data-driven information has been previously highlighted as a key point of vaccine promotion strategies [27]. Given the significant role of digital communication in public health, it is essential to improve how health professionals and coaches provide information to young athletes [6]. Tailored communication strategies that include popular digital platforms and provide accurate, engaging content could enhance vaccination rates among this group. Our findings highlight the need for a structured information program that addresses the specific concerns of young athletes. By understanding and utilizing the channels through which athletes get their information, health professionals can better promote the importance of influenza vaccination and other preventive measures. As of 2023, the National Greek Vaccination Guidelines suggest that people over 65 years of age and people with a high risk of infection or severe disease be vaccinated against influenza viruses [28]. However, there is no specific mention of people participating in sports. The importance of targeted vaccination guidelines in athletes has been highlighted in the past, yet most of the guidelines worldwide do not make any specific recommendations [29]. Considering the health risks an influenza infection can bring to athletes, their instructors and coaches in the National School of Physical education and Sport Science should urge the students (especially those participating in group sports) to get vaccinated. Health professionals could also visit the facilities of the National School of Physical education and Sport Science not only to provide information, but also to offer easy access to the vaccine. A recent review of the initiatives aimed at promoting influenza vaccination among the general population demonstrated that strategies involving interactive and integrated approaches were effective [30]. Comparable interventions targeting our target group may prove effective in enhancing the adoption of the flu vaccine.

### Limitations of This Study

Our results are subject to several limitations that need to be considered for their interpretation. To ensure the integrity of our study, it is important to acknowledge these limitations. Firstly, our analysis is based on a questionnaire survey that poses several possible biases. Considering that some individuals chose not to participate in our survey, the risk of non-response bias and self-selection bias should be taken into account, as the final sample may not be entirely representative of the whole population of young athletes. This potential lack of representativeness can affect the generalizability of our findings and should be considered when interpreting the results. Among the responders, response bias in the form of social desirability bias may have occurred. Additionally, we did not confirm the vaccination history of the participants, so the responses are entirely based on self-declaration. The number of professional athletes in our study was relatively small. If the sample size was larger, the results could potentially differ, reflecting different motivations and attitudes. Professional athletes might have unique concerns and motivations related to vaccination due to their intense training schedules, international travel, and higher needs in maintaining peak physical health. A larger sample could provide more robust data and insights into this subgroup, potentially highlighting distinct trends and considerations that are not fully captured in our current sample. Finally, the study period covers the period of the COVID-19 pandemic, which could have influenced the results. The pandemic has significantly impacted public health behaviors and attitudes towards vaccination. It is possible that increased awareness of and concerns about infectious diseases during this time led to increased interest in vaccinations, including the influenza vaccine. Conversely, the pandemic might have also led to vaccination fatigue or prioritization of the COVID-19 vaccine over the influenza vaccine. These factors could have influenced the respondents’ decisions and perceptions regarding influenza vaccination. We acknowledge this potential influence as a limitation factor that should be considered when interpreting the results of our study. Despite these limitations, many of the results of our analysis showed statistical significance, and we strongly believe that our findings provide valuable insight into the perception and the behavior of young athletes toward the influenza vaccine.

## 5. Conclusions

Our study aimed to bring to light the perception and behavior of young athletes toward the influenza vaccine. The analysis of the data revealed that only a minority of the study group (12.3%) had been vaccinated against the flu for the current period (2022–2023). Most of the participants attributed their non-vaccination status to the lack of time and the perception that a flu infection does not consist of a threat to their health. Despite these results, most of the participants deemed that the vaccines were safe and effective. Thus, they would suggest that their co-athletes get vaccinated. Similar to the findings of previous research, our data indicate that young athletes predominantly obtain their health information from non-scientific sources like social media and general media outlets. This reliance often leads to misinformation and contributes to vaccine hesitancy. Improving influenza vaccination rates among young athletes requires a multifaceted approach that includes enhancing information programs by health professionals and utilizing digital communication strategies effectively. By addressing the unique challenges and information sources of young athletes, we can promote better health outcomes and athletic performance.

## Figures and Tables

**Table 1 vaccines-12-00904-t001:** Demographics of the responders.

		N	%
Gender	Male	58	42
	Female	80	58
Level of training	Amateur level	82	59.4
	Semi-pro level	41	29.7
	Pro level	15	10.9
Age (years, mean, SD)		21.98 ± 3.96	
Sport activity category	Water sports	33	23.9
	Field sports	24	17.4
	Fighting sports	18	13
	Team sports	26	18.8
	Others	37	26.8
Source of information	Social media, media, and independent factors	100	72.5
	Scientific sources	38	27.5

**Table 2 vaccines-12-00904-t002:** Reasons for influenza non-vaccination.

Reasons	N	%
Lack of time	56	40.6
No fear of flu infection	50	36.2
Fear of vaccine side effects	10	7.2
The vaccine causing flu-like symptoms	5	3.6
Not willing to answer	17	12.3

**Table 3 vaccines-12-00904-t003:** Univariate analysis for flu vaccine uptake for season 2022–2023.

Variable	Flu Vaccination for 2022–2023		
	Yes (%)	No (%)	*p* Value
Sex			0.548
Male	6 (10.3)	52 (89.7)	
Female	11 (13.8)	69 (86.3)	
Age (Years, Mean, SD)	21.41 ± (4.84)	21.88 ± (3.84)	0.512
Level of sports			0.577
Amateur	12 (14.6)	70 (85.4)	
Semi-pro	4 (9.8)	37 (90.2)	
Pro	1 (6.7)	14 (93.3)	
Sports			0.266
Water sports	3 (9.1)	30 (90.9)	
Field sports	2 (8.3)	22 (91.7)	
Fighting sports	5 (27.8)	13 (72.2)	
Team sports	2 (7.7)	24 (92.3)	
Others (including tennis, dancing, weightlifting, and gymnastics)	5 (13.5)	32 (86.5)	
Source of information			0.853
Social media, media,	12 (12)	88 (88)	
and independent factors			
Scientific sources	5 (13.2)	33 (86.8)	
The vaccines are important for public health			0.179
Fully Agree/Agree	15 (11.5)	116 (88.5)	
Fully Disagree/Disagree	2 (28.6)	5 (71.4)	
In general, vaccines are safe.			0.022
Fully Agree/Agree	12 (9.9)	109 (90.1)	
Fully Disagree/Disagree	5 (29.4)	12 (70.6)	
In general, vaccines are effective.			0.081
Fully Agree/Agree	11 (9.9)	100 (90.1)	
Fully Disagree/Disagree	6 (22.2)	21 (77.8)	
Do you believe that you are going to lose training days due to the vaccine side effects?			0.117
Yes	7 (8.6)	74 (91.4)	
No	10 (17.5)	47 (82.5)	
Do you believe that you are going to lose an official game due to the vaccine side effects?			0.016
Yes	2 (3.8)	51 (96.2)	
No	15 (17.6)	70 (82.4)	

**Table 4 vaccines-12-00904-t004:** Univariate analysis for the flu vaccine in general.

Variable	Influenza Vaccination		
	Yes (%)	No (%)	*p* Value
Do you believe that you are going to lose training days due to the vaccine side effects?			0.001
Yes	20 (37.7)	33 (62.3)	
No	57 (67.1)	28 (32.9)	
Have you ever been infected with influenza?			0.002
Yes	49 (68.1)	23 (31.9)	
No	28 (42.4)	38 (57.6)	

## Data Availability

The study data are available from the corresponding author on reasonable request.

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
