# Peer review of "Influenza Vaccination Practices and Perceptions Among Young Athletes: A Cross-Sectional Study in Greece"

_vaccines, 2024, doi:10.3390/vaccines12080904_

Round 1

Reviewer 1 Report

Comments and Suggestions for Authors

This work deals with a very timely and current topic: vaccines hesitancy.

The study aims to explore the vaccination practices and intentions regarding influenza vaccines among young athletes through a structured, questionnaire-based to students of the National School of Sports in Greece.

Results show the importance of the absence of a well-organized information program from health professionals and coaches.

The article is very well organized well planned and methodologically sound. Results are, therefore conclusive.

Thus,  this reviewer is sure that this work has the potential to contribute to the wider literature on this topic.

Nonetheless, the study suffers from an important flaw that should be fixed prior to publication.

The Introduction goes straight to the point and does not offer an overview of the problem nor a theoretical framework.

In particular, results show the importance of the absence of a well-organized information program  from health professionals and coaches. But  the authors should discuss how young athletes get their information.

It is therefore seminal to discuss how athletes get their info.

First author should adrees the topic of how the public get info about the vaccines.

There is a wide literature that boomed durin Covid.

See:

Cervi, L.; Calvo, S.T.; Robledo-Dioses, K. Digital communication and the city. Analysis of the websites of the most visited cities in the world in the COVID-19 era. Rev. Lat. Comun. Soc. 2023, 81, 81–107 DOI:10.4185/RLCS-2023-1845

 After authors should discuss the specific case under scrutiny.

In addition the conclusion should 1) dialogue with previous studies to show similarities and differences 2) offer suggestions on how these info campaign should work.

Good luck!

Reviewer 2 Report

Comments and Suggestions for Authors

General comments.

Influenza vaccination hesitancy and access is a well-known and important public health problem. Specific population groups as sports athletes are a group of special study interest as risk to infection and vaccination behaviour and attitudes may differ from the general population. Therefore, the provided study article has certain added value to better understand the influence of vaccination hesitancy in this specific population, and this could also be interesting for the international reader.

The study has a descriptive character and uses a simple univariate analysis approach. An online interview survey among students from the School of Sports was used. It would be interesting to know if all the students were included in the sampling frame? How to explain that respondents (students) were subdivided into amateur, semi-professional, and professional athletes? More information about the school would be advisable. What is included in the term 'athletes' since most of the general population can participate in some kind of amateur sport?

Special comments and questions.

1.       The Section 'Introduction':

Lines 80 - 84: the idea about population groups who 'should be vaccinated' requires an explanation - is it recommended to be vaccinated? Vaccination covered financially? How about access to vaccination of those who are not in these groups (athletes): payments, who is doing vaccination, are there any obstacles? This can be related to the answer 'lack of time'?

2.       The section 'Results':

For tables 3 and 4 which differences P values are exactly measuring? The absolute numbers in the vaccinated and non-vaccinated groups differ markedlyis the P value for non-vaccinated or vaccinated?  Also, small numbers could be the reason for statistically nonsignificant difference – can this influence interpretation?

3.       The section 'Discussion':

As the study period covers the final or recent period of the Covidf-19 outbreak (pandemic), could the results be influenced by that?

Are those people the same, who are explaining nonvaccination with 'no time' and 'not important'?

As the number of professional athletes was small, could the results differ if their number is bigger (different motivation and attitudes)?  

Reviewer 3 Report

Comments and Suggestions for Authors

There are clear shortcomings of the study/design. The sample size is not big. Respondents come from the same background (the same school). While the authors cannot do anything about that, these should be acknowledged, and discussed/potentially mitigated. Some background info that would make it clear whether the people that participated were at least a representative sample of the 250 invitations or perhaps even greater population would be helpful.

  Methods were not described at all. The sentence "The  data were analyzed with SPSS software " is not really informative at all. The authors should explicitly state what kind of statistical tests were used.   Table 3 part about sports have the lines shifted   In the discussion, the authors should fit their findings within already known results. The authors say "Interestingly, our study demonstrated that the 169 main deterring factors seemed to be the lack of time". That did not seem too interesting to me as similar findings have been reported by other researchers for other kinds of vaccines for relatively benign diseases.  

Overall, the discussion is not truly a discussion, more like an extended summary of the results. In the discussion, the authors should compare their results with the results of other studies on influenza and similar diseases, either in Greece or other countries. Then they should discuss the similarities and differences between those studies and their results.

Round 2

Reviewer 1 Report

Comments and Suggestions for Authors

The manuscript has improved and is now ready to be published

Reviewer 3 Report

Comments and Suggestions for Authors

The authors adequately addressed all of my previous comments and the manuscript is now acceptable

I noticed a weird formatting on the title page with an extra 1 under the author's list, but that is just a typo not an issue that has any effect on scientific quality.